# Comparative iTRAQ proteomic profiling of sweet orange fruit on sensitive and tolerant rootstocks infected by 'Candidatus Liberibacter asiaticus'

Lixiao Yao[1,2], Qibin Yu[2], Ming Huang[2], Zhen Song[1], Jude Grosser[2], Shanchun Chen[1], Yu Wang[2], Frederick G. Gmitter, Jr.[2]*

**1** Citrus Research Institute, Southwest University, Chongqing, China, **2** Citrus Research and Education Center, University of Florida, Lake Alfred, Florida, United States of America

* fgmitter@ufl.edu

**Data Availability Statement:** All relevant data are within the manuscript and its Supporting Information files.

## Abstract

Citrus Huanglongbing (HLB), which is also known as citrus greening, is a destructive disease continuing to devastate citrus production worldwide. Although all citrus varieties can be infected with 'Candidatus Liberibacter asiaticus' (CaLas), a certain level of HLB tolerance of scion varieties can be conferred by some rootstocks. To understand the effects of rootstock varieties on orange fruit under CaLas stress, comparative iTRAQ proteomic profilings were conducted, using fruit from 'Valencia' sweet orange grafted on the sensitive ('Swingle') and tolerant rootstocks (a new selection called '46x20-04-48') infected by CaLas as experimental groups, and the same plant materials without CaLas infection as controls. The symptomatic fruit on 'Swingle' had 573 differentially-expressed (DE) proteins in comparison with their healthy fruit on the same rootstock, whereas the symptomatic fruit on '46x20-04-48' had 263 DE proteins. Many defense-associated proteins were down-regulated in the symptomatic fruit on 'Swingle' rootstock that were seldom detected in the symptomatic fruit on the '46x20-04-48' rootstock, especially the proteins involved in the jasmonate biosynthesis (AOC4), jasmonate signaling (ASK2, RUB1, SKP1, HSP70T-2, and HSP90.1), protein hydrolysis (RPN8A and RPT2a), and vesicle trafficking (SNAREs and Clathrin) pathways. Therefore, we predict that the down-regulated proteins involved in the jasmonate signaling pathway and vesicle trafficking are likely to be related to citrus sensitivity to the CaLas pathogen.

## Introduction

Huanglongbing (HLB), also known as citrus greening, is the most destructive citrus disease worldwide. It is commonly accepted that HLB is mainly caused by the gram-negative, phloem-limited fastidious bacterium 'Candidatus Liberibacter asiaticus' (CaLas) and vectored by *Diaphorina citri*, the Asian citrus psyllid. CaLas distributes in bark, leaf midrib, root, flower, and fruit parts of infected citrus trees, and a relatively high-concentration of bacteria can be

**Funding:** This work was supported by the Florida Citrus Research and Development Foundation (CRDF, Grant No. 15-010) (Frederick G. Gmitter Jr.), United States Department of Agriculture-National Institute of Food and Agriculture-Specialty Crop Research Initiative (USDA-NIFA SCRI, Grant No. 2017-70016-26328) (Frederick G. Gmitter Jr.), the National Key R&D Program of China (Grant No. 2018YFD0201500) (Zhen Song), the Earmarked Fund for China Agriculture Research System (CARS-27) (Shanchun Chen), and Fundamental Research Funds for the Central Universities (XDJK2019B018) (Lixiao Yao). The funders had no role in study design, data collection and analysis, decision to publish, or preparation of the manuscript.

**Competing interests:** The authors have decared that no competing interests exist

observed in fruit peduncles and fruit abscission zones [1, 2]. The *Ca*Las-infected trees usually display yellowing or blotchy mottled leaves, and show twig dieback, decline and even mortality several months to years after infection. Symptomatic fruits are usually found on symptomatic branches; they are lopsided and small, and often abscise prematurely. These fruits also have poor color development and may only "break color" on the stem end, the remaining surface of which is largely green. The abscission zone of the fruits located at the pedicel-fruit interface can be orange and their columella vascular bundles are orange or brown. It has been reported that juice from HLB-affected fruit was bitter and much more acidic with numerous off-flavors, similar to immature fruit [3]. This is because, comparing asymptomatic and healthy fruit, juice of symptomatic fruit contains lower percentage of soluble solid content and has lower soluble solid content to titratable acidity ratio [4, 5], and a significantly higher concentration of bitter limonoid compounds (limonin and nomilin) [5, 6]. Seeds are usually aborted in symptomatic fruit, and may be dark in color regardless if they are filled or partially filled.

Understanding molecular changes underlying host responses to pathogens is essential for clarification of the mechanisms behind plant-microbe interactions and in development of innovative strategies for both diagnostic and therapeutic approaches. Identifying key genes and proteins induced by HLB has been studied in leaf [7–20], stem [21], root [21, 22], fruit peel [2, 12, 23] with the "omic" methods, including microarray, RNA-Seq, 2-DE and iTRAQ. There is very little known about the proteomic differences in fruit pulp tissues as a consequence of HLB, and nothing is known on the impact of using more tolerant rootstocks on the fruit proteome. The materials utilized in these studies are varied, in general: 1) most of them focused on HLB-sensitive citrus [2, 7–14, 21–24], especially sweet orange 'Valencia', in which HLB-affected citrus were compared to healthy trees; 2) HLB-tolerant citrus [15], such as lemon, were also analyzed by comparing symptomatic leaves with asymptomatic leaves; and 3) the HLB-tolerant citrus and the HLB-sensitive citrus were analyzed at the same time to identify the genes/proteins associated with tolerance or sensitivity in citrus [16–20]. Overall, the results have shown the molecular mechanisms of disease development in HLB-tolerant and sensitive varieties are very complex, and within the tolerant citrus, different varieties showed different responses to HLB [16, 17].

The rootstock is very important for the commercialization of citrus trees and successful citrus production, as it can affect the citrus tolerance to abiotic and biotic stresses. At present, some rootstock varieties with HLB tolerance have been reported, but they have not yet been widely used in the commercial citrus [25]. Recent studies revealed that citrus trees grafted on some rootstock accessions showed tolerance to HLB. Some *Ca*Las-infected scion/tolerant rootstock trees continued to grow, and had significantly greater fruit load than the *Ca*Las-infected scion/sensitive rootstock trees [26–28]. Although Albrecht and her colleagues found that the concentrations of many metabolites were higher in the tolerant compared with sensitive rootstock cultivars [29], little is known about the molecular mechanism of rootstock effect on citrus scion response to *Ca*Las infection. Therefore, comparing scions grafted on tolerant and sensitive citrus rootstocks infected by *Ca*Las may provide more insights of the disease mechanisms.

*Ca*Las are unevenly distributed in the infected plants and could induce similar or different reactions among root, stem, leaf and fruit. As a sink tissue, HLB-associated characteristics in fruit are related to but not depending on restricted carbohydrate movement [2]. Due to widespread HLB within several years, citrus production in Florida was reduced sharply (Florida Citrus Commission: http://www.floridacitrus.org). The HLB-affected oranges also have been utilized in the juice processing and this can impact product flavor [3]. It is critical to understand the whole protein profile of the fruit, especially in fruit pulp, under the threat of HLB disease.

To explore the effects of *Ca*Las on the fruits harvested from 'Valencia' scion grafted on HLB-sensitive and HLB-tolerant rootstocks, we used an eight-channel iTRAQ technique to identify differentially expressed proteins by comparing symptomatic and healthy 'Valencia' fruits on the HLB-sensitive and HLB-tolerant rootstocks. Our results have highlighted the molecular processes regarding plant defense associated with HLB affected fruits produced from trees grown on rootstocks that differ in their sensitivity to the pathogen.

## Materials and methods

### Plant materials

Symptomatic and healthy 8-year-old 'Valencia' trees grafted on HLB-sensitive rootstock 'Swingle' (*Citrus paradisi* × *Poncirus trifoliata)* and HLB-tolerant rootstock '46x20-04-48' (*C. grandis* × *C. reticulata*) were grown at an orchard in St. Cloud, FL, USA, and were characterized by visual observation for symptom severity. Quantitative real-time PCR (qPCR) described by Li et al. [30] was used to determine the presence or absence of *Ca*Las in leaf and fruit tissues. Only two PCR-negative and healthy trees from each rootstock selection were found in the citrus grove, as a consequence of the widespread incursion on HLB in Florida when the study was initiated. Three biological replicate symptomatic trees were selected on each rootstock. Five fruits from each tree were randomly harvested and fruit pulp was collected from each fruit, mixed for one tree, and stored at -80°C. Other fruits were weighed, and their length and width were measured using a caliper. Minolta CR-330 colorimeter was used to measure peel color at three locations around the equatorial plane of fruit and the color was expressed as the Hunter ratio *a*/*b*. The color *a*/*b* ratio is negative for green, zero for yellow and positive for orange.

### The qPCR detection of *Ca*Las

Fruits and leaves were ground in liquid nitrogen using mortar and pestle. DNA extraction was performed using 100 mg of each ground tissue. Plant DNeasy Plant Mini Kit (Qiagen, Valencia, CA) was used to extract DNA according to the manufacturer's instructions, which yielded 25 ng of DNA per extraction. We performed qPCR assays using primers LAS_HLB1 (5´– `TC GAGCGCGTATGCAATACG –3´`) and LAS_HLB2 (5´– `GCGTTATCCCGTAGAAAAAGGT AG–3´`), and probe FAM_HLB (5´–`AGACGGGTGAGTAACGCG–3´`) [30]. Amplifications were performed using an Mx3005P qPCR System (Agilent Technologies, Santa Clara, CA) and the Brilliant III Ultra-Fast qPCR Master Mix (Agilent Technologies) according to the manufacturer's instructions. All reactions were carried out in two technical duplicates in a 10 μL reaction volume, 1 μL DNA as template per reaction. Leaf and fruit samples were considered PCR-positive if the Ct (cycle threshold) value was less than 32.

### Protein extraction

We used the modified phenol-based procedure to isolate total plant protein [31]. Specifically, five grams of fruit pulp frozen in liquid nitrogen was ground to powder and suspended directly in 15 mL of homogenization buffer (0.7 M Sucrose, 50 mM EDTA, 50 mM DTT, 1 mM PMSF, 0.1 M KCl, 1% 2-mercaptoethanol, 0.1% protease inhibitor mix, 0.5 M Tris [pH 7.5]) and the same volume of Tris-saturated phenol. The homogenate was vibrated and mixed vigorously for 30 min at 4°C. The top phenol phase was transferred into a new 50 mL polypropylene tube after 30 min centrifugation at 5000 ×*g* at 4°C. The protein pellet was precipitated overnight with 3 volumes of ice-cold 0.1 M ammonium acetate, and thoroughly washed four times with pure methanol and acetone. Air-dried protein was stored at -20°C.

## Proteomic iTRAQ running and data analysis

Ten protein samples were labeled and run for iTRAQ analysis (3 biological replicates of the symptomatic samples, 2 biological replicates of the healthy samples, and 2 rootstock varieties). 100 μg protein extracted with phenol from fruits on one tree was regarded as one biological replication. iTRAQ running and data analyses were performed by the Interdisciplinary Center for Biotechnology Research (ICBR) at the University of Florida (Gainesville, FL, USA). Protein digestion, marker labeling and cation exchange were performed according to the manufacturer's protocols. For protein identification, the MS/MS data produced in iTRAQ were analyzed by a thorough search against the NCBI subset of *C. sinensis* plants fasta database using the Paragon™ Algorithm of ProteinPilot V5.0 software suite (Applied Biosystems). For relative quantification of proteins, it was considered to be a differentially expressed (DE) protein if its *p* value was less than 0.05, and fold change was more than 1.5 (up-regulated), or less than 0.67 (down-regulated). The DE proteins were obtained between symptomatic fruit on 'Swingle' rootstock and healthy fruit on the same rootstock, and symptomatic fruit on '46x20-04-48' rootstock and healthy fruit on the same rootstock. Gene ontology analysis and functional classification of total DE proteins were performed using Blast2GO [32]. Arabidopsis orthologs were determined for each DE protein by local BLASTX (e-value $< 10^{-3}$) against the TAIR database of Arabidopsis predicted proteins. The biological interpretation of the DE proteins was further confirmed by MAPMAN [33] and Pathway Studio (Plant) Desktop 10 from Elsevier [34].

## Results

### HLB test by qPCR and the characteristics of fruits

To assess bacterial population levels in the 'Valencia' fruit and leaf tissues, qPCR analysis that targeted 16S rDNA of *Ca*Las was performed. The samples were labeled "positive" and "negative" if their Ct values were less and greater than 32, respectively. Regardless of rootstock varieties (HLB-tolerant '46x20-04-48' or HLB-sensitive 'Swingle'), leaf qPCR was positive and negative for the symptomatic and healthy trees, respectively. The fruits on healthy and symptomatic trees were marked as healthy or symptomatic fruits, respectively. The majority of fruits were found to be qPCR-negative. However, the symptomatic fruit from one tree on the tolerant rootstock '46x20-04-48' was positive (S1 Table).

For healthy fruit, there were no significant differences between the '46x20-04-48' and 'Swingle' rootstocks in terms of fruit width, fruit length, fresh weight and color (S2 Table). However, the Hunter ratio *a/b* of 'Valencia' fruit was negative for symptomatic fruit (green peel) under *Ca*Las infection. The size (width and length) and weight of symptomatic fruit were reduced 8% and 16%, and 21% and 40%, compared with healthy fruit from the trees grafted on the '46x20-04-48' and 'Swingle' rootstock, respectively.

### Fruit proteomic profile in symptomatic fruit on tolerant and sensitive rootstock

A total of 758 differentially-expressed proteins were found by comparing symptomatic fruit with healthy fruit, seven of which did not show orthology with any genes from *Arabidopsis thaliana*. There were 495 differentially-expressed fruit proteins (41 up, 454 down) found only in the symptomatic fruit on 'Swingle', whereas 185 proteins (6 up, 179 down) were found only in the symptomatic fruit on '46x20-04-48'. Fruit on both rootstocks shared 78 differentially-expressed proteins, all of which were down-regulated in fruit on the '46x20-04-48' rootstock, whereas seven were up-regulated in fruit on the 'Swingle' rootstock (Fig 1A). Most differentially-expressed proteins have catalytic activity and binding function, playing a role in

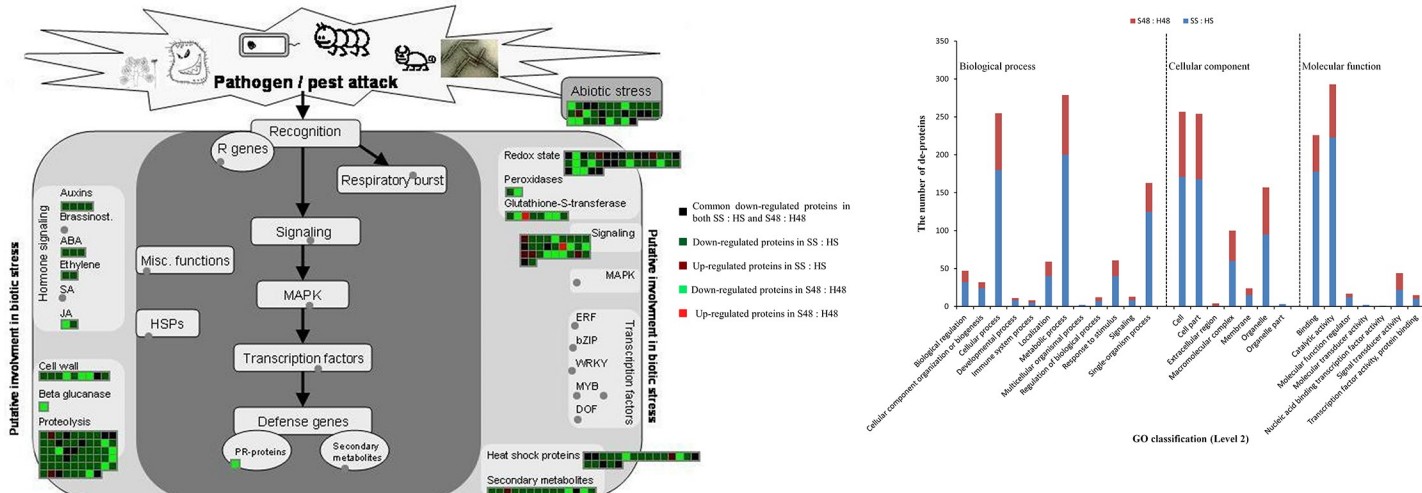

**Fig 1. Differentially-expressed fruit proteins in MAPMAN analysis and GO classification responding to Huanglongbing.** A: MAPMAN analysis; B: GO classification. H48 means healthy fruit from HLB-tolerant '46x20-04-48' rootstock; S48 means symptomatic fruit from HLB-tolerant '46x20-04-48' rootstock; HS means healthy fruit from HLB-sensitive 'Swingle' rootstock; SS means symptomatic fruit from HLB-sensitive 'Swingle' rootstock.

metabolic processes, single-organism processes, localization, and response to stimuli, etc. (Fig 1B). Fruit on 'Swingle' had more differentially-expressed proteins in GO classification, except in the extracellular category. There were no differentially-expressed proteins detected from trees grafted on '46x20-04-48' in the multicellular organismal process, molecular transducer activity, or nucleic acid binding transcription factor activity (Fig 1B).

## Differentially expressed transcription factors

There were nine transcription factors differentially-expressed by comparing the symptomatic and healthy fruits on the same rootstock type (Table 1). None of them was shared from symptomatic fruits on both 'Swingle' and '46x20-04-48' when compared with their healthy counterparts. Specifically, compared with healthy fruit from trees grafted on 'Swingle', symptomatic

**Table 1. The differentially-expressed transcription factors in Huanglongbing (HLB)-symptomatic fruit, from sensitive and tolerant rootstocks respectively.**

| Citrus ID | TAIR code[#] | Name | Description | SS: HS[a] | | S48: H48[b] | |
|---|---|---|---|---|---|---|---|
| | | | | Fold change | *P*-value | Fold change | *P*-value |
| gi\|641840770 | AT1G17880 | BTF3 | basal transcription factor 3 | | | 0.60 | 3.06E-02 |
| gi\|641849010 | AT2G20280 | F11A3.17 | zinc finger CCCH domain-containing protein 21 | 0.66 | 1.17 E-02 | | |
| gi\|641828464 | AT3G02790 | F13E7.27 | hypothetical protein | 0.62 | 3.28 E-02 | | |
| gi\|568833438 | AT2G02160 | F5O4.7 | zinc finger CCCH domain-containing protein 17 | 0.62 | 2.46 E-02 | | |
| gi\|641831608 | AT4G13850 | GR-RBP2 | glycine-rich RNA-binding protein 2 | 0.52 | 2.08 E-02 | | |
| gi\|641849752 | AT3G22830 | HSFA6B | heat stress transcription factor A-6b | 0.52 | 1.64 E-02 | | |
| gi\|641827820 | AT3G58680 | MBF1B | multiprotein-bridging factor 1b | 0.66 | 1.27 E-04 | | |
| gi\|641847522 | AT2G27100 | SE | SERRATE | 0.67 | 3.20 E-05 | | |
| gi\|641856847 | AT1G61730 | T13M11.9 | DNA-binding storekeeper protein-related transcriptional regulator | 2.07 | 3.44 E-03 | | |

[#]: TAIR means the Arabidopsis Information Resource

[a]: SS: HS mean symptomatic and healthy fruit on HLB-sensitive 'Swingle' rootstock.

[b]: S48: H48 mean symptomatic and healthy fruit on HLB-tolerant '46x20-04-48' rootstock.

fruit had one protein T13M11.9 (gi|641856847) that was up-regulated and seven other down-regulated proteins. Among these, HSFA6B (gi|641849752) was confirmed as response to heat stress, GR-RBP2 (gi|641831608) had glycine-rich domain, and F11A3.17 (gi|641849010), F5O4.7 (gi|568833438), and SE (gi|641847522) had the zinc finger domain. BTF3 (gi|641840770), a basal transcription factor, was the only differentially-expressed protein in symptomatic fruit on rootstock '46x20-04-48' compared with healthy fruit on '46x20-04-48'.

## Down-regulated proteins involved in hormone signal pathways

Compared with healthy fruit, there were 15 down-regulated proteins associated with the signaling pathway in symptomatic fruit (Fig 1A, S3 Table), six of which (AOC4, ASK2, APX1, CSD1, RUB1, and SKP1) were involved in the jasmonic acid, ethylene and salicylic acid cross-talk signaling pathway. Four (AHP5, PDV2, PPC1, and PPC4) and three proteins (ASK2, COP9, and SKP1) were involved in the cytokinins signaling and the senescence auxin pathways, respectively. Three (CDPK2, LTI65, and RAB18) and two (ASK2 and SKP1) proteins were involved in the ABA signaling and gibberellin signaling pathways. One protein (GLX1) was involved in the calcium signaling pathway. Compared with healthy fruit on 'Swingle', twelve down-regulated signal-associated proteins were found in the symptomatic fruit on the 'Swingle' rootstock. There were 4 down-regulated signal-associated proteins in the symptomatic fruit compared with the healthy fruit on the '46x20-04-48' rootstock.

Jasmonates play key roles in modulating defense responses and in regulating plant growth and development. Eleven down-regulated proteins involved in jasmonate biosynthesis and signal pathway were found in the symptomatic 'Valencia' fruits compared with healthy ones (Table 2). Two proteins involved in the jasmonate biosynthesis process were allene oxide cyclases (AOCs). In specific, AOC4 (gi|641805976) and AOC3 (gi|641851214) were detected with down-regulation in the symptomatic fruit on the 'Swingle' and '46x20-04-48' rootstocks, respectively. Eight down-regulated proteins involved in the jasmonate signal transmission were found in the symptomatic fruit on 'Swingle' rootstock and 4 down-regulated proteins in

**Table 2. Down-regulated fruit proteins involved in the jasmonate (JA) biosynthesis and signal pathway, in response to Huanglongbing (HLB).**

| Citrus ID | TAIR code[#] | Name | Description | SS: HS[a] | | S48: H48[b] | |
|---|---|---|---|---|---|---|---|
| | | | | Fold change | P-value | Fold change | P-value |
| JA biosynthesis | | | | | | | |
| gi|641805976 | AT1G13280 | AOC4 | Allene oxide cyclase 4 | 0.65 | 1.57 E-02 | | |
| gi|641851214 | AT3G25780 | AOC3 | Allene oxide cyclase 3 | | | 0.50 | 3.39 E-02 |
| JA signal pathway | | | | | | | |
| gi|641849213 | AT5G42190 | ASK2 | SKP1 1B-like | 0.59 | 2.46 E-03 | | |
| gi|641855847 | AT2G32120 | HSP70T-2 | Heat shock protein 70 | 0.51 | 3.25 E-02 | | |
| gi|568875065 | AT5G52640 | HSP90.1 | Heat shock protein 90 | 0.59 | 1.68 E-02 | 0.65 | 3.51 E-02 |
| gi|641837810 | AT1G31340 | RUB1 | Ubiquitin-NEDD8 | 0.56 | 3.11 E-02 | | |
| gi|568839457 | AT1G75950 | SKP1 | | 0.67 | 1.34 E-02 | 0.60 | 1.60 E-03 |
| gi|568834741 | AT4G19006 | AT4G19006 | 26S proteasome regulatory subunit | | | 0.54 | 1.79 E-02 |
| gi|568854739 | AT5G05780 | RPN8A | 26S proteasome non-ATPase regulatory subunit | 0.62 | 4.38 E-05 | | |
| gi|568853056 | AT4G29040 | RPT2a | 26S proteasome non-ATPase regulatory subunit | 0.62 | 1.12 E-02 | 0.65 | 3.00 E-04 |
| gi|568877704 | AT4G14110 | COP9 | COP9 signalosome complex subunit | 0.57 | 2.12 E-02 | | |

[#]: TAIR means the Arabidopsis Information Resource

[a]: SS: HS mean symptomatic and healthy fruit on HLB-sensitive 'Swingle' rootstock.

[b]: S48: H48 mean symptomatic and healthy fruit on HLB-tolerant '46x20-04-48' rootstock.

the diseased fruit on '46x20-04-48' rootstock. Among them, HSP90.1(gi|568875065), SKP1(gi|568839457), and RPT2a(gi|568853056) were found to be decreased in the symptomatic fruit, not only on the 'Swingle' but also on the '46x20-04-48' rootstock.

## Differentially-expressed proteins associated with vesicle trafficking

There were 16 differentially-expressed proteins involved in the vesicle trafficking process that were detected from the symptomatic fruit compared with their healthy counterparts (Table 3) on the 'Swingle' and '46x20-04-48' rootstocks. The expression of four soluble SNAREs (*N*-ethylmaleimide-sensitive factor attachment protein receptors) were decreased in the symptomatic fruit on the 'Swingle' rootstock. They are divided in the two subtypes that are t-SNAREs (gi|568850479 and gi|641843163) and v-SNAREs (gi|641840511 and gi|568822856) located on the target membrane and on the vesicular membrane, respectively. The t-SNARE protein gi|568850479 was also down-regulated in the symptomatic fruit on '46x20-04-48' rootstock. Expression of two other t-SNAREs (gi|641843247 and gi|641867568) were specifically decreased in the symptomatic fruit on the '46x20-04-48' rootstock. Four plastid division proteins were differentially expressed in the symptomatic fruit on 'Swingle', three of which were down-regulated (gi|568869681, gi|641835496, and gi|568852804). A clathrin protein (gi|568865883) and two clathrin interactors (EPSIN1 (gi|568844191) and EPSIN2 (gi|568850347)) were down-regulated in the symptomatic fruit on the 'Swingle' rootstock. Two proteins (gi|641830669 and gi|568845159), possibly providing energy for the vesicle trafficking, were down-regulated in the symptomatic fruit on 'Swingle'. The PTAC4 (gi|641826577) involved in the thylakoid formation was found in lower concentration in the symptomatic fruit on '46x20-04-48', but not in symptomatic fruit on 'Swingle'.

**Table 3. Differentially-regulated fruit proteins in vesicle trafficking responding to Huanglongbing (HLB).**

| Citrus ID | TAIR code[#] | Name | Description | SS: HS[a] | | S48: H48[b] | |
|---|---|---|---|---|---|---|---|
| | | | | Fold change | *P*-value | Fold change | *P*-value |
| gi\|568865883 | AT3G11130 | F11B9.30 | Clathrin | 0.63 | 3.11 E-02 | | |
| gi\|568850479 | AT5G16830 | SYP21 | Syntaxin 21 | 0.63 | 3.21 E-02 | 0.56 | 1.54 E-02 |
| gi\|641843247 | AT1G16240 | SYP51 | Syntaxin 51 | | | 0.61 | 5.92 E-04 |
| gi\|641843163 | AT1G28490 | SYP61 | Syntaxin 61 | 0.56 | 4.40 E-02 | | |
| gi\|641867568 | AT3G09740 | SYP71 | Syntaxin 71 | | | 0.62 | 7.17E-03 |
| gi\|641840511 | AT5G58060 | YKT61 | Similar to yeast SNARE YKT61 | 0.53 | 2.53 E-02 | | |
| gi\|568822856 | AT1G11890 | SEC22 | Secretion 22 | 0.59 | 1.36 E-02 | | |
| gi\|568844191 | AT5G11710 | AT5G11710 | Clathrin interactor EPSIN 1 | 0.61 | 4.90 E-03 | | |
| gi\|568850347 | AT2G43160 | F14B2.10 | Clathrin interactor EPSIN 2 | 0.57 | 2.00 E-03 | | |
| gi\|641830669 | AT2G44100 | GDI1 | Guanosine nucleotide diphosphate dissociation inhibitor 1 | 0.65 | 8.29 E-03 | | |
| gi\|568845159 | AT4G11150 | TUF | V-type proton ATPase subunit E1 | 0.64 | 5.23 E-04 | | |
| gi\|568869681 | AT2G28000 | Cpn60α | Chaperonin-60 alpha | 0.58 | 2.64 E-05 | | |
| gi\|641835496 | AT1G55490 | Cpn60β | Chaperonin 60 beta | 0.64 | 8.63 E-05 | | |
| gi\|568856420 | AT2G36250 | FtsZ2-1 | Cell division protein FtsZ homolog 2–1 | 1.57 | 3.11 E-02 | | |
| gi\|568852804 | AT2G16070 | PDV2 | Plastid division protein 2 | 0.58 | 2.04 E-03 | | |
| gi\|641826577 | AT1G65260 | PTAC4 | Plastid transcriptionally active 4 | | | 0.62 | 2.33E-05 |

[#]: TAIR means the Arabidopsis Information Resource

[a]: SS: HS mean symptomatic and healthy fruit on HLB-sensitive 'Swingle' rootstock.

[b]: S48: H48 mean symptomatic and healthy fruit on HLB-tolerant '46x20-04-48' rootstock.

## Other defense-response proteins

Despite the defense-response proteins described above, seventy-three others were also found with differential expression from comparing symptomatic and healthy fruits. They included 51 (3 up, 48 down) and 30 (all down) in symptomatic fruit on 'Swingle' and '46x20-04-48', respectively (S4 Table). Among these, eight were detected from the two varieties of rootstock under the HLB stress, most of which were decreased with expression in fruit on both 'Swingle' and '46x20-04-48' except gi|641841112 that was up-regulated and down-regulated in symptomatic fruit on 'Swingle' and '46x20-04-48', respectively. Two proteins (gi|568867630 and gi|641849727) involved in the PTI process and one protein (gi|641835987) involved in the ETI process were down-regulated in symptomatic fruit on 'Swingle' and '46x20-04-48', respectively. One PR-6 family protein (gi|568854491) was down-regulated in symptomatic fruit on the 'Swingle' rootstock. Arginase (gi|641867103), an acyl-CoA binding protein ACBP3 (gi|641862869), and phosphotase WIN2 (gi|641848484) were all down-regulated in symptomatic fruit on 'Swingle'. While, the polygalacturonase inhibitor 1 (PGIP1, gi|641820484) and a 6-phosphogluconolactonase EMB2024 (gi|568880520) were down-regulated in symptomatic fruit on rootstock '46x20-04-48' specifically.

## Discussion

HLB is a devastating disease affecting citrus worldwide. The leaf midrib is usually used to detect *Ca*Las by qPCR; however, the pathogen can be found in most HLB-affected fruit tissues including fruit pedicel, fruit abscission zone, fruit flavedo, vascular tissue, and juice vesicle [2]. The RNA transcription of HLB-impacted fruit versus girdled fruit comparison indicated that the mechanisms regulating development of HLB symptoms in fruit, rather than a direct consequence of carbohydrate starvation, are likely to be related to the host disease response [2]. In this experiment, both citrus trees grafted on the '46x20-04-48' and 'Swingle' rootstocks were naturally infected with *Ca*Las in the field. We identified more differentially-expressed proteins in symptomatic fruit on HLB-sensitive 'Swingle' rootstock (573) than HLB-tolerant '46x20-04-48' rootstock (263) compared with their healthy counterparts, which are similar to the published literatures [16, 17, 19] that more genes were differentially expressed in HLB-sensitive orange than the HLB-tolerant citrus. This phenomenon was also found on the metabolite products. The number of differentially regulated metabolites in the HLB-affected fruit was the largest in the susceptible cultivar and the lowest in the tolerant cultivar, compared with the non-affected fruit [29].

### Transcription factors were negatively regulated in the symptomatic fruit

We found nine down-regulated transcription factors in the symptomatic fruit on the *Ca*Las-infected tree compared with the healthy fruit on the non-pathogen-infected tree (Table 1), whose expression was associated with the biotic and abiotic stress. BTF3 (gi|641840770) protein, originally recognized as a basal transcription factor, was specifically down-regulated in symptomatic fruit on the '46x20-04-48' rootstock. It has been found that the gene *BTF3* from *Capsicum annuum* is involved in hypersensitive response cell death and regulates hypersensitive response-related gene expression [35]. gi|641831608 is a homologous protein of *GR-RBP2*, the gene of which encodes a glycine-rich RNA-binding protein that has RNA chaperone activity. *GR-RBP2* silencing was found to affect flowering time, stamen number, and seed development in Arabidopsis [36]. It is quite possible that the defect of this protein is associated with the aborted seeds of the HLB-affected citrus fruit. In addition, *GR-RBP2* overexpression can improve grain yield of rice (*Oryza sativa*) under drought stress conditions [37]. HSFA6B (gi|641849752) is a member of the heat stress transcription factor (Hsf) family and its cis-promoter elements can be bound with NAC019, which is a NAC transcription factor and its

expression can be induced by drought, high salt, and abscisic acid [38]. Multiprotein bridging factor 1 (MBF1) (gi|641827820), a transcriptional co-activator, functions in mediating transcriptional activation by connecting sequence-specific activator-like proteins and the TATA-box binding protein (TBP). Studies of maize and Arabidopsis reveal that MBF may be involved in stress response pathways [39, 40].

## Negative regulation of jasmonate signal in the symptomatic fruit

The plant hormone jasmonate is an important lipid-derived regulator responding to biotic and abiotic stresses as well as playing a role in plant growth and development. Its role in biosynthesis, perception, signal transduction, and action in stress response, growth, and development have been frequently discussed [41–43]. All the enzymes involved in the biosynthesis of jasmonate have been characterized from numerous plant species, which are also jasmonate-inducible. Allene oxide cyclase (AOC) is crucial within jasmonate biosynthesis, which establishes the enantiomeric structure of the cyclopentenone ring. AOC activity was determined by pairwise combinations of the four AOC isoforms and the most activitive heterodimers were found containing AOC4/AOC1 and AOC4/AOC2, respectively [44]. Proteins involved in jasmonate perception and signalling include COI1, acting as an F-box protein and jasmonate receptor, and the core compenent of an SKP1/Cullin/F-box complex (SCF$^{COI1}$) that functions as an E3 ubiquitin ligase. The SCF$^{COI1}$ that binds jasmonate make the transcriptional repressor JAZ to be poly-ubiquitylated, then the JAZ is subsequently degraded by the 26S proteasome [45]. The degradation of JAZ activities MYC2 and possible other transcription factors, allowing the expression of the plant systemic resistant genes [46].

COI1 also has been revealed to interact directly with the COP signalosome (CSN) in vivo using co-immunoprecipitation and gel-filtration analyses. More importantly, most of the *COI1*-dependent jasmonate-responsive genes also require CSN, whose abundance was shown to be important for jasmonate-dependant plant defense responses [47, 48]. At the same time, COI1 is a client protein of a chaperone complex SGT1b-HSP70-HSP90 and these complexes function in jasmonate hormone signalling through stabilizing the COI1 protein. The COI1 has direct interaction with SGT1b and is independent from SKP1 protein [49]. SKP1 has been demonstrated to interact with COI1 through yeast two-hybrid analysis and immunoprecipitation, and a SKP1-related protein, ASK2, has been also reported to interact with COI1 [50]. The Arabidopsis cullin AtCUL1, another subunit of SCF$^{COI1}$ E3 ubiquitin ligase, can be modified by the ubiquitin-related protein RUB1 [51]. The RUB family of proteins have around 50–60% of amino acids identical to ubiquitin. However, unlike ubiquitin modification, the conjugation product of RUB and Cullin does not appear to modify its metabolites' stability and stimulate SCF$^{COI1}$ ubiquitin ligase activity [52]. RUB1 interacts weakly with a proteasomal ubiquitin receptor Rpn10, but binds to proteasome ubiquitin-shuttle proteins like ubiquitin [53].

In the symptomatic fruit from trees grafted on the 'Swingle' rootstock infected with *Ca*Las, the citrus homologous proteins involved in the jasmonate biosynthesis and signal pathway were accumulated at significantly lower levels compared with the healthy fruit on the same rootstock variety. The jasmonate biosynthesis enzyme AOC4, that could compose the highest AOC activity heterodimers, was down-regulated 35% in the HLB-affected fruit on 'Swingle' [44]. The fruit proteins involved in the jasmonate signal pathway (SKP1, RUB1, HSP90.1, HSP70T-2, ASK2, and COP9) were down-regulated 31% - 49% in the citrus tree grafted on 'Swingle' infected with *Ca*Las. The subunits of 26S proteasome, RPB8A and RPT2a, were decreased to 38% expression. Recently, Zhong et al. [22] have found 5 *JAZ* genes were up-regulated in the *Ca*Las-infected citrus root through RNA-seq, which might inhibit MYC expression. These down-regulated and up-regulated proteins indicated the plant resistance induced

by jasmonate was suppressed, and were likely to be related to facilitate *Ca*Las infection on citrus as shown in Fig 2.

### Negative regulation of vesicle trafficking in the symptomatic fruit

Plants lack an acquired immune system and a circulatory system. They have evolved conserved innate immune systems against pathogenic microbes on two definable layers. PTI is the immunity

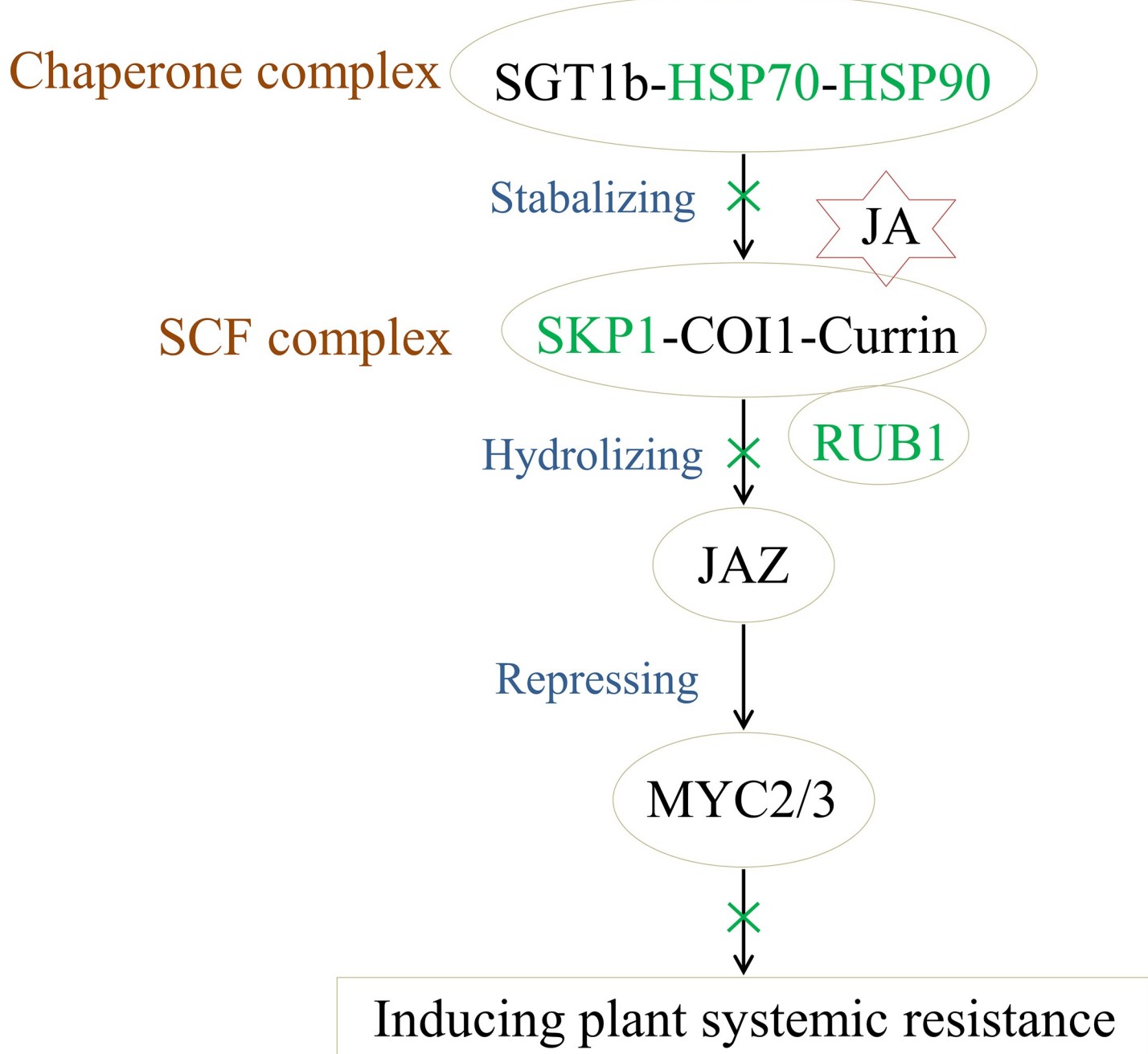

**Fig 2. Scheme of JA signal pathway in the symptomatic fruit responding to Huanglongbing.** These green proteins were down-regulated in symptomatic fruit on the HLB-sensitive rootstock 'Swingle'. This suggested perhaps a negative effect, depending on COI1, suppressing the hydrolyzation of JAZ and interfering with the expression of plant resistance genes.

triggered with pathogen-associated molecular patterns, and ETI is triggered with pathogen-associated effectors [54]. A variety of host proteins involved in PTI and ETI are largely limited in the distinct membrane subcellular structures of plant cells. In order to activate the spatio-temporally controlled immune responses, it is critical to regulate the dynamic subcellular localization of plant immunity-related proteins through the engaged protein trafficking system [55, 56].

There are a number of proteins required to play proper roles together from vesicle budding to vesicle fusion to ensure the correct vesicle trafficking, among which, SNARE polypeptides are the most important that function during the recognition and fusion between vesicle and target membrane [57]. They are divided into the two functional forms: 1) v-SNAREs that are inserted into the vesicular membrane, and 2) t-SNAREs that are located on target membrane. Generally, through forming a *cis*-SNARE complex located on the target membrane, t-SNARE proteins (2~3) properly interact with a v-SNARE to form a four-helix *trans*-SNARE complex on an incoming vesicle via their coiled-coil domains that provides fusion specificity [58, 59]. In symptomatic fruit on the 'Swingle' rootstock, we found two t-SNAREs (SYP21 and SYP61) and two v-SNAREs (YKT61 and SEC22) were down-regulated under the *Ca*Las infection (Table 3). The v-SNARE SEC22 has a subcellular distribution compatible with a role at the ER-Golgi interface. The overexpression of SEC22 and Memb11 can induce collapse of Golgi membrane proteins collapsing into the ER, which are also involved in anterograde protein trafficking at the ER-Golgi interface [60]. These down-regulated t-SNAREs and v-SNAREs in symptomatic fruit on the 'Swingle' rootstock are likely to affect vesicle trafficking and subsequently weaken the plant immune defense.

Clathrin-mediated endocytosis enters cells through clathrin-coated vesicles on the plasma membrane. Clathrin plays a major role in modifying membranes to form the budding vesicle. Rather than directly binding to the membrane or to cargo receptors, it relies on adaptor and accessory proteins (such as EPSIN) that are recruited to the plasma membrane [61]. Clathrin protein (gi|568865883), EPSIN1 (gi|568844191) and EPSIN2 (gi|568850347) were down-regulated in symptomatic fruit on the 'Swingle' rootstock (Table 3). In Arabidopsis, EPSIN1 interacting with clathrin and v-SNARE VTI11 plays a crutial role in the vacuolar trafficking of soluble proteins at the *trans*-Golgi network [62] and EPSIN2 is essential in protein trafficking through interacting with clathrin and v-SNARE VTI12 and phosphatidylinositol-3-P [63].

Two distinct plant plastid chaperonin polypeptides, Cpn60α (gi|568869681) and Cpn60β (gi|641835496), were down-regulated in symptomatic fruit on the 'Swingle' rootstock. In Arabidopsis, *len1* mutant showed that Cpn60β (LEN1) functions as a molecular chaperone in chloroplasts and its deletion triggers cell death, which leads to the establishment of systemic acquired resistance [64]. The formation of a normal plastid division apparatus requires Cpn60α and Cpn60β. A proper level of Cpn60 is required for folding of stromal plastid division proteins and/or the regulation of FtsZ polymer dynamics [65]. The distribution of active GTP and inactive GDP-bound forms between membranes and cytosol is controlled by GDI (gi|641830669), a key regulator of Rab/Ypt GTPases. A RabGDI deletion is lethal in yeast [66]. TUF (gi|568845159) is the subunit E1 of the vacuolar type $H^+$-ATPase (v-ATPase), and v-ATPase localizes to specific regions of the vacuolar membrane and functions to generate a proteon motive force for secondary transport systems [67]. The two proteins, GDI and TUF, possibly provide energy for the vesicle trafficking and were down-regulated in symptomatic fruit on the HLB-sensitive rootstock 'Swingle'.

## Other down-regulated plant defense proteins in the symptomatic fruit

Many down-regulated proteins in the symptomatic fruit on the HLB-sensitve 'Swingle' rootstock were shown to be involved in the plant defense response according to GO annotation.

Arginase (gi|641867103), producing ornithine and urea, plays a role in plant defense responses. In tomato, inducing arginase activity in leaves has been shown to respond to wounding, treatment with jasmonate and infection with *Botrytis cinerea*, *Plasmodiophora brassicae*, and *Pseudomonas syringae* [68–70]. ACBP3 (gi|641862869), present in the extracellar space, can transport fatty acid/lipid precursors. It plays a role in the plant defense, responding to the bacterial pathogen *P. syringae* pv *tomato* DC3000 [71], which is likely to serve as one of the potential pathogen targets, and its degradation can enhance pathogen growth [72]. Interacting with the bacterial HopW1-1 effector, WIN2 (gi|641848484) (a PP2C) is required against virulent *P. syringae* [73]. PGIPs (gi|641820484), capable to recognize and directly inhibit fungal endopolygalacturonases, are extracellular leucine-rich repeat (LRR) proteins present in the plant cell wall [74]. Constitutive expression of *Vvpgip1* from *Vitis vinifera* was found to protect tobacco plants from *Botrytis cinerea* through remodelling and reorganizing the cellulose xyloglucan network in cell walls [75, 76]. EMB2024 (gi|568880520), known as PGL3, is a 6-phosphogluconolactonase that catalyses a reaction step of the pentose phosphate pathway in plastids. Knockdown of PGL3 constitutively activated the plant salicylic acid-dependent defense responses [77].

## Conclusions

'Valencia' scions show differences in the fruit characteristics and protein expression profiles under *Ca*Las infection when grafted on HLB-sensitive 'Swingle' and HLB-tolerant '46x20-04-48' rootstocks. Most differentially-expressed fruit proteins were down-regulated in the symptomatic fruit on both 'Swingle' rootstock (525 down-regulated proteins) and '46x20-04-48' rootstock (257 down-regulated proteins) compared with their healthy counterparts. The down-regulated proteins in the jasmonate signal pathway and vesicle trafficking are likely to affect the plant defense response that enhance citrus sensitivity to *Ca*Las, the HLB pathogen. Further research to unravel the underlying mechanisms through which the rootstock interacts with the scion to enhance scion tolerance of HLB can lend valuable insight to the disease manifestation processes, potentially leading to new genetic or even cultural approaches to decrease the serious negative impacts of HLB on citrus production globally.

## Supporting information

**S1 Table. Detection of *Ca*Las 16S rDNA from leaf and fruit tissues of 'Valencia' by qPCR.**
(XLSX)

**S2 Table. The characteristics in healthy and symptomatic fruit grafted on a tolerant ('46x20-04-48') and sensitive ('Swingle') rootstock.**
(XLSX)

**S3 Table. Differentially-expressed proteins associated with plant signaling in the symptomatic fruit.**
(XLSX)

**S4 Table. Other differentially-expressed defense-response proteins in the symptomatic fruit.**
(XLSX)

## Author Contributions

**Data curation:** Lixiao Yao, Qibin Yu, Zhen Song.

**Formal analysis:** Lixiao Yao, Ming Huang.

**Funding acquisition:** Lixiao Yao, Zhen Song, Shanchun Chen, Frederick G. Gmitter, Jr.

**Methodology:** Ming Huang, Yu Wang.

**Resources:** Jude Grosser.

**Supervision:** Frederick G. Gmitter, Jr.

**Writing – original draft:** Lixiao Yao, Qibin Yu, Shanchun Chen, Yu Wang.

**Writing – review & editing:** Qibin Yu, Frederick G. Gmitter, Jr.

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
