## [Decision Letter · Decision Letter 0]

2 Jan 2020

PONE-D-19-33708

Comparative iTRAQ proteomic profiling of sweet orange fruit on sensitive and tolerant rootstocks infected by ‘Candidatus Liberibacter asiaticus’

PLOS ONE

Dear Dr. Yao,

Thank you for submitting your manuscript to PLOS ONE. After careful consideration, we feel that it has merit but does not fully meet PLOS ONE’s publication criteria as it currently stands. Therefore, we invite you to submit a revised version of the manuscript that addresses the points raised during the review process.

We would appreciate receiving your revised manuscript by Feb 16 2020 11:59PM. To enhance the reproducibility of your results, we recommend that if applicable you deposit your laboratory protocols in protocols.io, where a protocol can be assigned its own identifier (DOI) such that it can be cited independently in the future. For instructions see: http://journals.plos.org/plosone/s/submission-guidelines#loc-laboratory-protocols

We look forward to receiving your revised manuscript.

Kind regards,

Yuan Huang

Academic Editor

PLOS ONE

Reviewers' comments:

Reviewer's Responses to Questions

**Comments to the Author**

1. Is the manuscript technically sound, and do the data support the conclusions?

Reviewer #1: Yes

Reviewer #2: Yes

2. Has the statistical analysis been performed appropriately and rigorously? 

Reviewer #1: Yes

Reviewer #2: Yes

3. Have the authors made all data underlying the findings in their manuscript fully available?

Reviewer #1: Yes

Reviewer #2: Yes

4. Is the manuscript presented in an intelligible fashion and written in standard English?

Reviewer #1: Yes

Reviewer #2: Yes

5. Review Comments to the Author

Reviewer #1: Through comparing proteomics of citrus fruit from cLas infected and uninfected Valencia sweet orange scions grafted on Huanglongbing susceptible and tolerant rootstocks, this study identified different proteomic alterations after cLas infection in a same scion grafted on two different types of rootstocks, which could possibly explain the better quality and yield of fruit grafted on tolerant rootstocks. The design of this study and selection of materials were reasonable, and the methods applied should be robust enough for publication. But there were still some small problems including imprecise description and grammar mistakes in the manuscript, so I suggest minor revision.

Line 24 ‘incurable’ is not accurate. In some studies and reports HLB affected trees could be cured by applying antibiotics or high temperature treatment. The main challenge of HLB is that there has not been any robust controlling method that could solve the problem once for all.

Line 36 and 39 should be ‘jasmonate signaling’

Line 47 delete ‘economically’

Line 49 ‘non-culturable’ is controversial and should be deleted, some study has reported the culture of the bacteria though at very low titer

Line 54 What does ‘non-specific’ mean? Do other disease have similar symptoms? If yes why it was still ‘readily distinguishable’? Should be explained or modified.

Line 55-56 Do symptomatic fruits always abscise prematurely? If not, ‘often’ should be added in front of abscise

Line 62-63 ‘fruit contains lower percentage of soluble solid content and has lower soluble solid content to titratable acidity ratio’

Line 63 ‘amount of’ should not followed by non-countable noun

Line 64, lack an ‘and’

Line 71 ‘most of them’

Line 78 delete ‘phenomena’ and ‘even’, and use past form ‘showed’

Line 82 “yet they have not yet been”

Line 85 delete ‘even’, should denote whether the HLB-sensitive scion/rootstock is CaLas-infected

Line 89 ‘tolerant and sensitive citrus rootstocks’

Line 91 Better modified to be ‘are unevenly distributed’ and ‘could induce similar or different’

Line 93 ‘Due to spreading quickly and the difficulties to control HLB’ should be rewritten

Line 95-97, ‘Because’ and ‘therefore’ used together. There seemed to be no strong causal relationship here, and sentence had better be re-structured

Line 163 ‘characteristics’

Line 176 Has statistical analysis been done to test the size and weight different? Are there any DE proteins related with fruit quality alteration? There is no discussion on this area at all, then why choosing fruit rather than leaves or other tissues as experimental material.

Line 178 ‘proteomic profile’

Line 182-183 Different numbers of DE proteins from that described in Abstract

Line 263 ‘serious destructive’ has grammar error. ‘uncured disease’ not appropriate here

Fig 1. The go terms had better be put in the graph instead of the number IDs which would make the graph easier to read.

There are still a few other grammar mistakes should be revised.

Reviewer #2: Huanglongbing is the most serious citrus disease in the world. The manuscript entitled “Comparative iTRAQ proteomic profiling of sweet orange fruit on sensitive and tolerant rootstocks infected by ‘Candidatus Liberibacter asiaticus’” is an interesting article. The authors found a very interesting phenomenon that fruit of susceptible citrus scions can gain the ability of HLB tolerance from their rootstocks. To figure out the underlying mechanism, the authors employed strategy of comparative proteomic analysis and finally found out some potential factors and pathways. In my opinion, this manuscript is well written and provide some novelty results to this research field. However, some questions should be clarified to reader before publication.

Minor revision:

As shown, the current tables in the manuscript only presented a small part of the differentially expressed proteins, and all other data were enclosed in the supplementary, that restrict the readers to get a full view of the result. A table is fine to present a small number of proteins, but quite limited for hundreds of proteins. Since abundant of differentially expressed proteins were obtained from the comparative proteomic analysis, a more convenient and intuitive interface is quite necessary to present your data, such as pathways combined with hot-map.

It is better to use only the name of rootstock instead of “HLB tolerant” or “HLB sensitive” throughout the manuscript after the first introduction of material background.

Line 28-31: Confused long sentence. Short sentence can be more understandable, e.g. "same materials without CaLas infection were used as controls”.

After revision, this manuscript could be publish in this journal.

6. PLOS authors have the option to publish the peer review history of their article (what does this mean?). If published, this will include your full peer review and any attached files.

Reviewer #1: No

Reviewer #2: No

---

## [Author Response · Author response to Decision Letter 0]

23 Jan 2020

For Reviewer #1 

Suggestion Line 24 ‘incurable’ is not accurate. In some studies and reports HLB affected trees could be cured by applying antibiotics or high temperature treatment. The main challenge of HLB is that there has not been any robust controlling method that could solve the problem once for all.

Reply: We replaced “incurable’ with ‘destructive’ in line 23 of “Manuscript_PONE-D-19-33708”.

Suggestion Line 36 and 39 should be ‘jasmonate signaling’; Line 47 delete ‘economically’; Line 49 ‘non-culturable’ is controversial and should be deleted, some study has reported the culture of the bacteria though at very low titer.

Reply: Have done in lines 35, 37, and 44 of “Manuscript_PONE-D-19-33708”; ‘non-culturable’ was replaced with more commonly recognized terminology ‘fastidious’ in line 46.

Suggestion Line 54 What does ‘non-specific’ mean? Do other disease have similar symptoms? If yes why it was still ‘readily distinguishable’? Should be explained or modified.

Reply: We deleted the “non-specific” sentence, for the following sentences described the characteristics of symptomatic fruit.

Suggestion Line 55-56 Do symptomatic fruits always abscise prematurely? If not, ‘often’ should be added in front of abscise; Line 62-63 ‘fruit contains lower percentage of soluble solid content and has lower soluble solid content to titratable acidity ratio’.

Reply: Have done.

Suggestion Line 63 ‘amount of’ should not followed by non-countable noun

Reply: Change ‘amount of’ to ‘concentration of’.

Suggestion Line 64, lack an ‘and’; Line 71 ‘most of them’; Line 78 delete ‘phenomena’ and ‘even’, and use past form ‘showed’; Line 82 “yet they have not yet been”; Line 85 delete ‘even’, should denote whether the HLB-sensitive scion/rootstock is CaLas-infected; Line 89 ‘tolerant and sensitive citrus rootstocks’; Line 91 Better modified to be ‘are unevenly distributed’ and ‘could induce similar or different’

Reply: Have done.

Suggestion Line 93 ‘Due to spreading quickly and the difficulties to control HLB’ should be rewritten; Line 95-97, ‘Because’ and ‘therefore’ used together. There seemed to be no strong causal relationship here, and sentence had better be re-structured

Reply: We rewrote it.

Suggestion Line 163 ‘characteristics’

Reply: Have done in line 156 of “Manuscript_PONE-D-19-33708”.

Suggestion Line 176 Has statistical analysis been done to test the size and weight different? Are there any DE proteins related with fruit quality alteration? There is no discussion on this area at all, then why choosing fruit rather than leaves or other tissues as experimental material.

Reply: We did perform statistical analyses to test the size and weight differences; the results are mentioned in lines 165-170 of “Manuscript_PONE-D-19-33708”. Just like the leaves and roots, the development and maturity of fruit is under the HLB stress. Many studies have shown the differentially expressed genes or proteins in leaves and roots. However, we did not know if there are CaLas-sensitive or –tolerant pathways involved in fruits. This is the reason for choosing fruit as the experimental material. We added this sentence to the introduction: “There is very little known about the proteomic differences in fruit pulp tissues as a consequence of HLB, and nothing is known on the impact of using more tolerant rootstocks on the fruit proteome”.

Suggestion Line 178 ‘proteomic profile’

Reply: Have done in line 171 of “Manuscript_PONE-D-19-33708”.

Suggestion Line 182-183 Different numbers of DE proteins from that described in Abstract

Reply: We checked the data carefully and corrected the numbers of DE proteins. For example, we said 573 DE proteins in the symptomatic fruit on ‘Swingle’ in Abstract part, which equaled with 495 DE fruit proteins found only in the symptomatic fruit on ‘Swingle’ and 78 DE proteins shared in fruit on both rootstocks in Results part.

Suggestion Line 263 ‘serious destructive’ has grammar error. ‘uncured disease’ not appropriate here

Reply: We rewrote it.

Suggestion Fig 1. The go terms had better be put in the graph instead of the number IDs which would make the graph easier to read.

Reply: Have done.

Suggestion There are still a few other grammar mistakes should be revised.

Reply: We did our best to correct the other grammar mistakes.

For Reviewer #2

Suggestion As shown, the current tables in the manuscript only presented a small part of the differentially expressed proteins, and all other data were enclosed in the supplementary, that restrict the readers to get a full view of the result. A table is fine to present a small number of proteins, but quite limited for hundreds of proteins. Since abundant of differentially expressed proteins were obtained from the comparative proteomic analysis, a more convenient and intuitive interface is quite necessary to present your data, such as pathways combined with hot-map.

Reply: We used MapMan to analyze the differentially expressed proteins in biotic stress. The new figure (Figure 1A) provides a more comprehensive view of the results.

Suggestion It is better to use only the name of rootstock instead of “HLB tolerant” or “HLB sensitive” throughout the manuscript after the first introduction of material background.

Reply: Have done in most places, though kept this designation in the Conclusion section for clarity.

Suggestion Line 28-31: Confused long sentence. Short sentence can be more understandable, e.g. "same materials without CaLas infection were used as controls”.

Reply: Have done.

---

## [Editor Report · Decision Letter 1]

27 Jan 2020

Comparative iTRAQ proteomic profiling of sweet orange fruit on sensitive and tolerant rootstocks infected by ‘Candidatus Liberibacter asiaticus’

PONE-D-19-33708R1

Dear Dr. Yao,

We are pleased to inform you that your manuscript has been judged scientifically suitable for publication and will be formally accepted for publication once it complies with all outstanding technical requirements.

With kind regards,

Yuan Huang

Academic Editor

PLOS ONE
---

## [Editor Report · Acceptance letter]

5 Feb 2020

PONE-D-19-33708R1 

Comparative iTRAQ proteomic profiling of sweet orange fruit on sensitive and tolerant rootstocks infected by ‘<I>Candidatus</I> Liberibacter asiaticus’ 

Dear Dr. Yao:

I am pleased to inform you that your manuscript has been deemed suitable for publication in PLOS ONE. Congratulations! Your manuscript is now with our production department. 

With kind regards,

on behalf of

Dr. Yuan Huang 

Academic Editor

PLOS ONE